# Fibrostenosing Crohn’s Disease: Pathogenetic Mechanisms and New Therapeutic Horizons

**DOI:** 10.3390/ijms25126326

**Published:** 2024-06-07

**Authors:** Irene Mignini, Valentina Blasi, Fabrizio Termite, Giorgio Esposto, Raffaele Borriello, Lucrezia Laterza, Franco Scaldaferri, Maria Elena Ainora, Antonio Gasbarrini, Maria Assunta Zocco

**Affiliations:** CEMAD Digestive Diseases Center, Fondazione Policlinico Universitario “A. Gemelli” IRCCS, Università Cattolica del Sacro Cuore, Largo A. Gemelli 8, 00168 Rome, Italy; irene.mignini@guest.policlinicogemelli.it (I.M.); v.blasi.97@gmail.com (V.B.); giorgio.esposto2@gmail.com (G.E.); raffaeleborr@gmail.com (R.B.); lucrezia.laterza@policlinicogemelli.it (L.L.); franco.scaldaferri@policlinicogemelli.it (F.S.); mariaelena.ainora@policlinicogemelli.it (M.E.A.); antonio.gasbarrini@unicatt.it (A.G.)

**Keywords:** Crohn’s disease, intestinal fibrosis, anti-fibrotic treatments

## Abstract

Bowel strictures are well recognized as one of the most severe complications in Crohn’s disease, with variable impacts on the prognosis and often needing surgical or endoscopic treatment. Distinguishing inflammatory strictures from fibrotic ones is of primary importance due to the different therapeutic approaches required. Indeed, to better understand the pathogenesis of fibrosis, it is crucial to investigate molecular processes involving genetic factors, cytokines, alteration of the intestinal barrier, and epithelial and endothelial damage, leading to an increase in extracellular matrix synthesis, which ultimately ends in fibrosis. In such a complex mechanism, the gut microbiota also seems to play a role. A better comprehension of molecular processes underlying bowel fibrosis, in addition to radiological and histopathological findings, has led to the identification of high-risk patients for personalized follow-up and testing of new therapies, primarily in preclinical models, targeting specific pathways involving Transforming Growth Factor-β, interleukins, extracellular matrix balance, and gut microbiota. Our review aims to summarize current evidence about molecular factors involved in intestinal fibrosis’ pathogenesis, paving the way for potential diagnostic biomarkers or anti-fibrotic treatments for stricturing Crohn’s disease.

## 1. Introduction

Crohn’s disease (CD) is a chronic inflammatory bowel disease (IBD) that may affect any district of the gastrointestinal (GI) tract, from the upper GI segments to the anus. It shows a high degree of heterogeneity in clinical manifestations and natural history, leading to the need for patients’ categorization. A classification based on three main features (age at diagnosis, location, and clinical behavior) was first proposed in 1998 during the World Congress of Gastroenterology in Vienna [1] and subsequently updated in Montreal in 2005 [2]. According to behavior, CD may be classified as inflammatory (non-stricturing and non-penetrating), structuring, or penetrating, eventually associated with perianal localization [2]. Disease patterns may change over time; thus, while the large majority of patients present at the beginning with inflammatory CD, the development of stricturing or penetrating complications is observed in about 45% of cases 10 years after diagnosis [3].

Stenoses in CD are the result of inflammatory processes or fibrosis accumulation within the intestinal wall, or, more frequently, a mix of both phenomena. Indeed, chronic inflammation drives the excessive production of extracellular matrix (ECM), ultimately leading to fibrosis development [4]. As a consequence of these mechanisms, the fibrostenosing phenotype is characterized by persistent luminal narrowing and can result in dangerous complications, notably bowel obstruction. Although strictures can affect any segment of the GI tract, they most frequently concern the small bowels, reflecting the distribution of inflammation, while colonic stenoses are less common and more often related to malignancy occurrence [5]. Consistently, ileal CD is more often stricturing, while fistulas are more frequently described in colonic or ileo-colonic localization [3]. At diagnosis, about 11% of patients present with bowel strictures [3], a percentage that tends to increase over time due to disease progression, reaching about 15% at 10 years and 22% at 20 years [6]. The typical symptoms and signs of stricturing CD are abdominal distension, pain, absence of gas/stool passage, and vomiting. Progressive narrowing of the intestinal lumen ends in bowel obstruction, which represents, together with fistulas, the main indication for surgery in CD [4].

The management of stenoses in CD still represents a major challenge for IBD physicians. When bowel narrowing is mainly attributable to active inflammatory disease, treatment with anti-inflammatory drugs (conventional or biologic therapy) may be attempted [7]. However, despite the huge and quick expansion of the pharmacological armamentarium for CD in the last few years, all the currently available drugs primarily address inflammation, and their effect in preventing fibrostenosing complications is scarce. To date, no specific molecules to reduce or reverse fibrosis have been approved [7]. Thus, after fibrotic progression, treatment still relies on surgical or endoscopic approaches. Acute bowel obstruction, a potential life-threatening condition, requires emergency intervention in the presence of ischemia, perforation, or peritonitis [8]. In milder cases, guidelines from the European Crohn’s and Colitis Organisation (ECCO) recommend primary conservative treatment with fasting, intravenous (IV) fluids, and gastric decompression if needed, followed by elective surgery, while the use of IV corticosteroids is limited to stenoses with a predominant inflammatory component [7,8]. When surgery is required, both intestinal resection and strictureplasty are indicated, with the latter being preferred in patients with long segments of affected bowels. Endoscopic balloon dilatation is a safe and effective option suitable for short strictures (<5 cm) and an alternative to surgery, depending on operator expertise and patient preference [7,8]. The therapeutic algorithm for bowel obstruction, based on ECCO recommendations, is represented in Figure 1. 

Considering the strong impact of intestinal strictures on patients’ quality of life and prognosis, great scientific effort has been made in the last few decades to improve the management of this complication of CD. Aiming to provide clinicians with practical guidance, ECCO published in 2016 a topical review on fibrostenosing CD [7]. Moreover, two years later, a multidisciplinary panel of experts (the CrOhN’S disease anti-fibrotic STRICTure Therapies group—CONSTRICT) was established with the purpose of addressing the unmet needs, and a consensus was developed to standardize definitions and assessments of stricturing CD, creating a framework for future clinical trials [9]. These papers present an updated overview of diagnostic methods, fibrosis predictors, and the management of fibrostenosing CD. 

However, many issues remain unanswered, notably the definition of the most appropriate diagnostic techniques, the quantification of inflammatory and fibrotic components within the stenosis, the lack of reliable biomarkers, and effective anti-fibrotic drugs. In such a context, it becomes crucial to improve our understanding of fibrosis etiopathogenesis, trying to identify the molecular mechanisms that may prove useful both as fibrosis biomarkers and therapeutic targets.

In our review, we aim to depict the complex landscape of intestinal fibrosis pathogenesis, summarizing the state of the art about the molecular and cellular pathways involved. First, we thoroughly analyze studies investigating the role of genetic and epigenetic factors, cytokines and soluble mediators, ECM components, mesenteric fat, and gut microbiota in promoting or preventing fibrogenic processes. We then describe how such knowledge may be applied for therapeutic purposes, highlighting the limited impact of the approved drugs on fibrostenosing CD and examining the pioneering data about some new potential medical treatments. 

This is a narrative, not a systematic review, and it may be encumbered by some biases in article selection due to the lack of a structured search strategy. However, the main focus of our work is on the molecular bases of fibrostenosing CD in order to stress how important a deep comprehension of etiopathogenetic mechanisms is for the clinical management of the disease. Data on molecular pathways are largely derived from pre-clinical studies, showing such wide variability that a systematic analysis of the current literature is hardly challenging. Therefore, the ultimate purpose of our paper is to provide a general framework for future research, pursuing the identification of therapeutic targets and paving the way for novel clinical trials on human beings. In the text, the terms “stenosis” and “stricture”, as well as “stricturing CD” and “fibrostenosing CD”, are used as synonyms.

## 2. Pathogenetic Mechanisms of Intestinal Fibrosis in CD 

In CD, chronic inflammation leads to repeated events of injury to the bowel wall and subsequent repair, ultimately resulting in fibrosis development. The process of fibrogenesis is characterized by an abnormal accumulation of ECM rich in collagen, which is produced in response to different stimuli by various types of cells, including smooth muscle cells, myofibroblasts, and fibroblasts [4]. Mechanisms influencing the activation of these cells involve genetic and epigenetic factors, microbiota alterations, cytokines, and other soluble factors released during inflammatory processes [10,11,12]. Modulating all these targets could offer the possibility of developing anti-fibrotic therapies acting at different levels. The next subsections address the different specific mechanisms involved in intestinal fibrogenesis, which are schematically summarized in Figure 2.

### 2.1. Genetic Factors

The role of multiple genes in the pathogenesis of intestinal fibrosis in CD has been investigated, potentially contributing to the high familiarity of the disease. Genome-wide association studies have identified many common variants in CD patients, but the current data about genetic susceptibility are largely heterogeneous [13]. The most relevant genes include nucleotide-binding oligomerization domain-containing gene 2 (NOD2), interleukin-23 receptor (IL23-R), autophagy-related 16-like 1 gene (ATG16L1), C-X3-C Motif Chemokine Receptor 1 (CX3CR1), and transforming growth factor beta (TGF-β) [14]. Their role in fibrogenesis is represented in Figure 3. Other genes include matrix metalloproteinases-3 (MMP-3), membrane-associated guanyl-ate kinase (MAGI1), Janus kinase 2 (JAK2), fucosyltransferase 2 (FUT2), interleukine-12 beta (IL12β), immunity-related GTPase M (IRGM), and unc-51-like autophagy-activating kinase 1 (ULK1) [15].

Undoubtedly, the gene with the most recognized influence on CD pathogenesis is NOD2, a susceptibility gene related to the risk of disease occurrence. It is expressed in dendritic cells, granulocytes, T-cells, and terminal ileal Paneth cells [16], and it encodes for an intracellular pattern recognition receptor that recognizes muramyl dipeptide (MDP) [17], a component of bacterial peptidoglycan that is involved in activating the innate immune system in CD [15,18]. Upon binding to MDP, NOD2 undergoes oligomerization and associates with the scaffolding kinase RIP2. RIP2, in turn, forms oligomers, facilitating the direct transmission of NOD2’s signal to the IKK complex. This process culminates in the activation of the nuclear factor (NF)-kB signaling pathway [19]. In addition, NOD2 is involved in two more processes: the activation of the NLR3 inflammasome, which activates caspase-1, converting pro-IL1β into its active form [20], and the stimulation of autophagy through ATG16L1 [21]. Three NOD2 polymorphisms situated in the LRR region are directly correlated with CD. Among these, the most recognized is the frameshift mutation (L1007fs), while the other two are missense mutations (R702W and G908R) [22,23]. It is hypothesized that these variants may impair the function of the LRR domain, affecting its ability to recognize microbial components and/or inhibit NOD2 dimerization. Consequently, this leads to the inappropriate activation of NF-κB in monocytes, hindering bacterial clearance and causing intestinal permeability issues [23,24,25]. The crucial role of NOD2 has been recognized for several years. In 2011, Adler et al. published a meta-analysis including 49 studies and 8893 subjects with CD, finding that the presence of a single NOD2 mutation increased the risk of an aggressive phenotype (stricturing or penetrating) by 8%, while an increase of 41% was calculated in the case of two mutations of the gene. The presence of two mutations predicted a complicated disease with a specificity of 98% [26]. A more recent meta-analysis by Dang et al. further supports the involvement of NOD2 in severe CD phenotypes: by analyzing 28 studies (14 of which are about NOD2), the authors confirm NOD2 as the main risk allele for CD recurrence after surgery [27]. Moreover, variants are associated with increased levels of TGF-β1 and collagen and are well known to correlate with the fibrostenosing phenotype in CD [28]. 

IL-23R encodes a vital subunit of the receptor for the pro-inflammatory cytokine IL-23. This receptor is abundantly expressed on memory T cells, monocytes, natural killer cells, and dendritic cells. The IL23R-STAT3 pathway mediates pro-inflammatory responses by driving the production of IL-17 through T helper 17 (Th17) lymphocyte activation [29]. Following the identification of IL-23R as a susceptibility gene for CD [30], a study exploring the relationship between the rs1004819 SNP within IL-23R and disease features was conducted, observing a heightened occurrence of ileal involvement and fibrostenotic disease among individuals homozygous for the TT allele compared to those with the CC wild-type allele. However, these associations did not remain significant after the statistical correction was applied [31]. Specific polymorphisms of the same allele rs1004819, as well as the allele rs7517847, also seem to be associated with stricturing/penetrating behavior in a Polish and Bosnian cohort analyzed in a study by Borecki et al. [32].

The ATG16L1 gene encodes a protein crucial in the autophagosome pathway, which is pivotal for targeting and eliminating pathogen-derived proteins in the innate immune response as well as degrading worn cytoplasmic components [29,33,34]. The ATG16L1 T300A variant (rs2241880) has been identified as a susceptibility factor for CD [35,36]. An impairment of ATG16L1 can lead to an increase in the unfolded protein response (UPR), causing heightened endoplasmic reticulum stress and an enhanced NF-kB pathway [37]. Consequently, this results in a reduction in pathogen clearance and an imbalance in cytokine production. Additionally, the presence of this ATG16L1 variant appears to correlate with a reduced capability to generate a specific type of macrophage (referred to as Mφind), with a distinct anti-inflammatory function [38]. Consequently, the resulting inflammatory signals may stimulate mesenchymal cells to produce large quantities of collagen and other fibrogenic molecules. The ATG16L1 T300A variant has been demonstrated to be associated with ileal disease [34,39] and fibrostenotic disease [39], irrespective of NOD2 status or disease duration. However, the latest findings from the European IBD chip Project (a European Commission-funded multicenter effort aimed at defining the influence of genetic factors on disease phenotype and response to therapies in IBD patients) did not confirm the link between ATG16L1 T300A and fibrostenotic disease [40].

CX3CR1 serves as a receptor involved in leukocyte chemotaxis and adhesion. It binds fractalkine, a chemokine expressed by epithelial and endothelial cells, with characteristics of both traditional chemokines and adhesion molecules. The expression of CX3CR1 is detected mainly on CD8+ T cells. Through its interaction with fractalkine, it regulates the migration of CD8+ intraepithelial lymphocytes into the intestinal lamina propria [41]. Following bacterial stimulation, CX3CR1-expressing cells adhere swiftly to inflamed vascular endothelium, potentially facilitating the passage of cytotoxic effector cells into the pathogenic milieu [42]. Brand et al. examined two SNPs (V249I, rs3732379, and T280M, rs3732378) in the context of CD, finding an association between both SNPs and fibrostenotic disease, though not independently of ileocolonic disease location [41]. Sabate et al. observed a similar trend toward fibrostenotic behavior in V249I carriers (but not in T280M carriers), especially among smokers, regardless of NOD2 L1007fs carriage and ileal involvement [42]. Importantly, functional studies revealed that peripheral blood mononuclear cells (PBMCs) from individuals with wild-type CX3CR1 genotypes exhibit stronger adhesion to membrane-bound fractalkine compared to those from homozygous V249I–T280M donors, emphasizing a tangible role of the CX3CR1/fractalkine axis in CD fibrosis [43].

Finally, while the role of TGF-β in fibrosis development is widely recognized, the influence of its polymorphisms on genetic susceptibility for bowel stenoses is still controversial, and some authors have questioned the impact of TGF-β polymorphisms on stricturing CD [44,45]. As far as we know, the only study demonstrating an association between TGF-β variants and the fibrostenosing phenotype was published in 2006 by Hume et al. In this case–control study, they performed a genotype—phenotype analysis for both TGF-β1 codon 25 and angiotensinogen-6, considering two cohorts of IBD patients (both CD and UC) and CD families together with controls. They described an association between a polymorphism of TGF-β1 codon 25 (Arg/Arg) and the fibrostenosing phenotype in CD with a faster progression to surgery. In regards to angiotensin II, it is thought that it can increase the production of TGF-β1; however, the authors found no association between disease phenotype and promoter polymorphism of the angiotensinogen-6 GRA [46]. Such results suggest that other factors are probably involved in this process, for which further studies are needed.

### 2.2. Epigenetic Factors

Chronic inflammation in CD can alter gene expression through epigenetic regulation, thus further contributing to the development of intestinal fibrosis [47]. Epigenetic modifications related to the development of fibrosis include alterations in micro-ribonucleic acid (miRNA), histone modification, and deoxyribonucleic acid (DNA) methylation. 

miRNAs are endogenous non-coding RNAs made up of a few nucleotides (around 20–30) that participate in the process of epigenetic regulation by interacting with the target messenger RNA (mRNA). Starting with their production, they are incorporated into the RNA-induced silencing complex (RISC) and induce gene silencing by overlapping with complementary sequences that are present on target mRNA molecules. This binding results in the repression of translation or degradation of the target molecule [48,49]. Alterations in the expression of miRNAs have been found in various pathogenetic processes relating to inflammation, autoimmunity, and carcinogenesis [50,51,52]. Different miRNA expression profiles were observed both in the serum and within the intestinal mucosa between patients with active and inactive inflammatory disease. In the myofibroblast and muscle cells of CD patients, miRNA influenced the expression of TGF-β signaling, resulting in the continuous activation of TGF-β1 and, consequently, the excessive production of collagen and ECM [53]. In a study by Nijhuis et al., a reduction in the expression of miRNAs belonging to the miR-29 family was demonstrated at the level of intestinal strictures compared to adjacent areas with normal caliber. In this study, cultures of intestinal fibroblasts isolated from patients with CD were analyzed. It was demonstrated that the inhibition of miR-29b expression induced by TGF-β was necessary for TGF-β to stimulate collagen synthesis, favoring the fibrosis process. From this study, it emerges that low levels of miR-29 may have a profibrotic role at the level of stenoses in CD [54]. Other studies had previously investigated the association between renal, hepatic, and cardiac fibrosis and the expression of miR-29 [55,56,57]. Furthermore, miR-29 is implicated in the genetic regulation of molecules involved in the composition of the ECM, including collagen (COL1, COL3, and COL4) [58]. It seems that NOD-2 is one of the factors that can regulate the expression of miR-29. The presence of NOD-2 gene variants is associated with the risk of developing a fibrostenosing phenotype in CD, and this mechanism also appears to involve the inhibition of miR-29 expression [59]. In a study by Lewis et al., low levels of miRNA-19 were also associated with stricturing CD [60]. Similarly, miRNA-21 proved to be involved in CD pathogenesis by interfering with the PI3K/Akt/mTOR pathway (phosphatidylinositol 3-kinase/protein kinase B/mammalian target of rapamycin), a pro-inflammatory and pro-fibrotic signaling pattern, which is further described above. Wang et al. recently demonstrated that miRNA-21 decreased in the intestinal mucosa of CD patients compared with healthy controls, and low levels of miRNA-21 resulted in an upregulation of the PI3K/Akt/mTOR pathway [61]. Interestingly, the same group focused, in another study, on the role of miRNA-21 in fibrostenosing CD, finding that it was upregulated in tissues from fibrotic bowels in comparison to non-fibrotic ones, and inhibiting miRNA-21 reduced EMT [62]. 

Also, histone modification represents another level of epigenetic regulation. Intestinal fibrosis and endothelial–mesenchymal transition (EndoMT) seemed to be influenced by the action of cytokines on chromatin, which is related to the expression of the COL1A2 gene. The most abundant component of the ECM is type I collagen, made up of a heterotrimer of two α-1 chains and one α-2 chain (COL1A2). Sadler et al. demonstrated how some profibrotic cytokines (IL-1β, Tumor Necrosis Factor alpha—TNF-α—, and TGF-β) are able to determine histone modifications (hypermethylation of histone 3 and histone 4 hyperacetylation) and phosphorylation of RNA polymerase at the COL1A2 promoter. An interesting fact observed in the study was that after the removal of the cytokine stimulus (16 days of exposure), the acetylation of lysine 16 in histone 4 of the promoter region persisted [63]. These findings suggest a central role of chromatin modifications in the regulation of gene expression in EndoMT and intestinal fibrosis.

DNA methylation seems to be a consequence of the inflammatory process and determines altered gene expression in intestinal fibroblasts [47,64]. The relationship between inflammation and DNA methylation was studied by Sominemi et al., who analyzed blood samples from a pediatric population with CD compared with healthy controls. Their study revealed a different pattern of DNA methylation on CpG islands between CD patients and healthy controls. Alterations in methylation are correlated with plasma levels of the C-reactive protein. Furthermore, during patients’ follow-up, they observed that, after treatment, DNA methylation patterns of CD patients were more similar to healthy control profiles, regardless of the progression from inflammatory to stenosing disease phenotype [64]. Instead, Sadler et al. analyzed fibroblasts from the colon of patients with fibrostenosing CD compared with healthy controls. The study revealed that the major differences in DNA methylation profiles between CD and controls were not found at CpG islands but at the level of introns and intergenic regions. In particular, three genes were identified (PTGDS, PTGIS, and WNT2B) whose DNA methylation in the promoter region inversely correlated with gene expression in fibrotic fibroblasts [47]. Ahmad et al. analyzed DNA methylation in mucosal tissues from CD patients requiring and not requiring surgery. In both groups, they compared specimens from healthy mucosa with ones from affected mucosa. Interestingly, they found not only differential DNA methylation between the two groups of patients, but also unique methylation signatures in healthy vs. diseased tissue. Globally, their study suggests an evolution of the methylation profile (the “methylome”) from early to advanced CD stages [65]. 

### 2.3. Cytokines and Other Soluble Factors

Chronic inflammation in CD leads to the activation of immune and non-immune cells, which release pro-fibrogenic mediators targeting ECM-producing cells [66]. Macrophages, neutrophils, basophils, mast cells, monocytes, and eosinophils produce a large amount of pro-fibrotic and pro-inflammatory molecules, notably TNF-α, TGF-β1, and different interleukins (IL-1β, IL-4, IL-6, and IL-13). Immune cells include T-helper cells (Th1, Th2, and Th17), regulatory T cells (Tregs), and B cells, which all exert different functions. Th1 signaling involves interferon gamma (INF-γ) and seems to have antifibrotic activity. Conversely, Th2 releases IL-4, IL-5, and IL-13, showing profibrotic activity. The Th17 pathway is both proinflammatory and profibrotic, while Treg probably inhibits Th17 and Th2-driven fibrosis [67].

Different cytokines and fibrotic growth factors exert their functions in the ECM [68,69]. Among the fibrotic growth factors, a predominant role in fibrogenesis is undoubtedly played by TGF-β, as well as in CD and in fibrotic diseases affecting other organs than the GI tract [70,71]. 

TGF-β is a regulatory protein that has three isoforms (TGF-β1, TGF-β2, and TGF-β3), of which the first one is the most common. TGF-β is secreted as a precursor bound to a propeptide and subsequently cleaved in the Golgi apparatus. From here, it is transported into the ECM, linked to a further peptide, and activated by various molecules, including MMPs, integrins, thrombospondin-1, reactive oxygen species (ROS), and bone morphogenetic 1 (BMP-1) [72]. Once activated, TGF-β binds to its receptor on the cell surface, leading to the phosphorylation of intracellular proteins, including small mother against decapentaplegic (SMAD) [73]. As a consequence, the SMAD pathway is activated with translocation into the nucleus of the phosphorylated SMAD2/3 complex and SMAD4 and subsequent regulation of the target genes [74,75]. The final effect of this process is the stimulation of ECM deposition with the development of fibrosis through the activation of Wingless-Int-1 (Wnt)-β-catenin signaling [76,77]. At the same time, TGF-β1 activates SMAD-inhibiting proteins (SMAD6 and SMAD7) that compete with the SMAD receptor and induce degradation of the TGF-β receptor [78].

The role of some other interleukins in the development of fibrosis has also been investigated. For instance, IL-6 is involved in an autocrine pathway that stimulates fibroblasts to produce cell migration-inducing hyaluronan-binding proteins, which is increased in patients with CD. The consequence is hyaluronan degradation, which contributes to the maintenance of inflammation and fibrosis [79]. Another cytokine implicated in the pathogenesis of fibrosis appears to be IL-36, which consists of three forms of agonists (IL-36α, IL-36β, and IL-36γ) and a receptor antagonist (IL-36Ra), all binding to the same receptor complex (IL-36R) and finally recruiting the accessory protein IL-1RAcP [80]. IL-36’s role in fibrosis development is largely described in organs other than the gut, such as the lung, skin, kidney, and heart [81,82]. In the last few years, its involvement in bowel diseases has also been hypothesized [83]. Scheibe et al. analyzed the expression of IL-36 and its receptor (IL-36R) in intestinal tissue samples from patients with IBD (92 with CD and 48 with UC) compared to controls. Their results revealed that subjects with fibrostenosing CD had increased tissue levels of IL-36, which correlated with an increased number of activated fibroblasts, producing α-smooth muscle actin (α-SMA), a profibrogenic factor [84]. More recently, the same group demonstrated that IL-36 signaling interferes with the expression of the metalloproteinase MMP13. They observed that MMP13 is increased both in gut specimens from patients with stenotic CD and in mouse models with dextran sulfate sodium (DSS)-induced colitis. The intraperitoneal injection of IL-36R agonists enhanced the expression of MMP13 [85].

Lastly, the role of the PI3K/Akt/mTOR pathway is also a matter of study. It is a highly conserved signal network promoting cell survival and growth; therefore, it is largely investigated in the oncologic field [86]. Its influence on CD pathogenesis has also been hypothesized. Long et al. found that such a pathway was upregulated in peripheral CD4+ T cells of CD patients compared with healthy controls [87]. Consistently, the inhibitors of mTOR (mTORis) are considered antifibrotic soluble factors, as they are able to reduce the synthesis of collagen and the number of fibroblasts and myofibroblasts. They inhibit the production of profibrogenic cytokines such as IL-4, IL-6, IL-13, IL-17, and TGF-β1 [88]. 

Table 1 schematically summarizes the main role of the aforementioned cytokines and their final protective or promoting role in the fibrogenesis process.

### 2.4. Creeping Fat

The hypertrophy of the mesenteric adipose tissue, also called creeping fat, was first described in 1952 by Crohn et al. as one of the major characteristics of CD [89]. It represents a source of proinflammatory and profibrotic cytokines, participating in the deposition of the ECM and the process of intestinal fibrogenesis [90]. Many cytokines involved in the process of fibrogenesis (IL-6, IL-12, IFN-γ, and TGF-β) are responsible for the activation of proinflammatory cascades that induce CD4+ T cells to differentiate in Th17 and Th2, thus perpetuating inflammation and driving the fibrogenic process. Cytokine is increased by modifications in mesenteric fat, including the invasion of inflammatory cells and specific bacteria [91]. Moreover, in response to cytokines, the expansion of adipose tissue and progression to creeping fat have been described [92,93].

### 2.5. Extracellular Matrix

The process of ECM deposition is quite complex, and it depends on the balance between ECM production and degradation. In the bowel, many cells may become ECM-producing myofibroblasts, including not only resident mesenchymal cells like smooth muscle cells, subepithelial myofibroblasts, and fibroblasts [94,95], but also bone marrow stem cells, pericytes, epithelial, or endothelial cells through epithelial mesenchymal transition (EMT) or endoMT [96,97,98], circulating precursor fibrocytes [99], and intestinal stellate cells [100]. 

In CD, the expression of pro-fibrotic cytokines and ECM molecules correlates with the presence of fibroblast islands in bowel strictures [101,102]. This results in the thickening of intestinal wall layers with the expansion of muscularis mucosa and muscularis propria [103,104] and fibromuscular obliteration of the submucosa, with the accumulation of collagen bands and smooth muscle cells, especially in the small bowel [105,106].

On the other side, the apoptosis of myofibroblasts is crucial for maintaining tissue homeostasis and granting fibrosis resolution. Among the various genes involved in this process are NOD2 and ATG16L1, which are expressed by myofibroblasts, proven to have a proapoptotic effect through the activation of caspases [107], and well-known proteases for their primary importance in regulating apoptosis [108]. Such evidence further supports the association of variants of these genes with a higher risk of fibrostenosing CD. Also, hepatocyte growth factor (HGF) seems to be able to stimulate myofibroblast apoptosis through MMPs with an antifibrotic effect [14]. MMPs are endopeptidases that are produced in response to inflammatory stimuli by many cells (macrophages, monocytes, neutrophils, T-cells, and mesenchymal cells) [109]. They are secreted as inactive zymogens and activated through proteolytic cleavage. Once activated, they degrade collagens, fibronectins, and laminins in the ECM. Some factors, like tissue inhibitors of metalloproteinases (TIMPs) and α2-macroglobulin, modulate the activity of MMPs [110]. Consistently, TIMPs, by inhibiting MMPs, suppress myofibroblast apoptosis [107]. Patients with IBD show an altered balance between MMPs and TIMPs. This results in an exaggerated production of ECM mediated by MMPs and reduced degradation of ECM mediated by TIMPs [109,110].

### 2.6. Gut Microbiota

The gut microbiota also seems to play a role in the development of intestinal fibrosis [26,111]. As pointed out in a recent review by Bernardi et al. focusing specifically on this topic, different microbial metabolites have been proven to be involved both in intestinal inflammation and fibrosis occurrence [112]. One of the mechanisms contributing to the pathogenesis of IBD is the loss of tolerance to resident intestinal bacteria, which leads to an exaggerated immune response in genetically predisposed individuals [113]. In patients with CD, an imbalance between protective and harmful bacteria has been found, resulting in dysbiosis [114]. A greater relative abundance of proteobacteria and a lower relative abundance of firmicutes were observed in CD patients [115].

If the role of the microbiota in the pathogenesis of the disease has been investigated in several studies, less evidence is available regarding the role of the microbiota in the process of fibrogenesis. Some studies have highlighted possible associations between microbiota modulation and the fibrogenesis process, also hypothesizing the involvement of gastrointestinal infections. A study by Grass et al. demonstrated that *Salmonella enterica serovar Typhimurium* infection causes a profibrotic response in the gastrointestinal tract of mice. In this study, intestinal tissue samples from mice infected with *Salmonella enterica serovar Typhimurium* were analyzed. Immunohistochemical analysis was performed to evaluate the expression of profibrotic cytokines. The study highlighted how *Salmonella* infection, which resulted in chronic colonization of the cecum and colon, was associated with the development of intestinal fibrosis characterized by the presence of fibroblasts and myofibroblasts and the deposition of collagen type I in the mucosa, submucosa, and muscularis mucosae, with high levels of profibrotic cytokines [116]. Other studies have shown that some microbial components influence the differentiation of mesenchymal cells with an increased proliferation rate of myofibroblasts in IBD patients [117]. 

The role of the microbiota in modulating the fibrosis process induced by cytokines, such as tumor necrosis factor-like cytokine 1A (TL1A), was also investigated. TL1A is a cytokine belonging to the TNF family that is expressed by fibroblasts and implicated in the process of inflammation and fibrosis in IBD [118]. In mice, induced TL1A overexpression was associated with the development of spontaneous ileitis and the worsening of fibrosis. A study by Jacob et al. demonstrated that overexpression of the TL1A factor in mice without resident microbial flora was not associated with the development of a proinflammatory and profibrotic phenotype, suggesting the role of the microbiota in promoting this process [119].

As highlighted above, CD patients with one or more mutations in the NOD2 gene have an increased risk of fibrosis development. Maeda et al. showed that mice with homozygous NOD21007fs exposed to MDP present a higher activation of the nuclear factor kappa-light-chain enhancer of activated B cells (NF-κB), increasing inflammation and alterations of the intestinal barrier with bacterial translocation [120]. Moreover, studies on murine models show that, in sterile conditions, knockout NOD2 mice do not develop colitis spontaneously but only after bacterial introduction, suggesting a potential role of a trigger from the microbiome in this process [121]. Indeed, in a study by Zhao et al., theTLR5 ligand flagellin demonstrated that it could trigger the fibrotic phenotype of intestinal fibroblasts through a post-transcriptionally regulated process [122]. Flagellins represent a component of the bacterial surface that influences motility and adhesion [123] but also interacts with TLR5, consequently activating NF-kB, and leading to the production of proinflammatory cytokines [124]. 

## 3. Anti-Fibrotic Therapies

To answer the urgent need for anti-fibrotic therapies for CD, the Stenosis Therapy and Anti-Fibrotic Therapy (STAR) consortium has been created [125]. In the last few years, it has made a great effort to improve the management of stricturing CD, identify standardized scoring systems for intestinal fibrosis, and define common therapeutic endpoints for use in clinical trials and everyday practice [125,126,127]. Interestingly, research about fibrotic diseases affecting organs other than the gut has progressed much quicker, especially concerning the liver, lung, and skin. Many randomized clinical trials have been conducted, testing a wide range of molecules, notably intracellular enzymes and kinases or agents modulating growth factors, inflammatory processes, ECM formation, the renin-angiotensin system, or 5-hydroxy-3-methylglutaryl-coenzyme A (HMG-CoA) reductase [128,129,130,131]. In a recent review, the STAR consortium highlighted how such notable experience from other diseases may also be useful for stricturing CD, suggesting that gastroenterologists should learn from other medical areas to address the open issues about intestinal fibrosis [132]. 

Indeed, the scientific literature concerning the gastrointestinal field has recently shown a growing interest in new potential anti-fibrotic agents targeting molecules or cells involved in the fibrogenic process at different levels, but the available evidence is mainly derived from pre-clinical studies and needs further confirmation.

In the following subsections, we first summarize evidence about the effect on fibrostenosing CD of currently approved drugs, whose main mechanism of action remains anti-inflammatory. Subsequently, we provide a general overview of the most relevant progress on specific anti-fibrotic treatments that are still under development, as well as the currently ongoing clinical trials testing new potential therapeutic targets. Finally, in a specific subsection, we highlight issues that are still open, paving the way for future research. 

### 3.1. Approved Systemic Medical Therapies

As mentioned before, current approved therapies for CD mainly address inflammation and may exert a positive effect on CD stenoses with a predominant inflammatory component. Corticosteroids rapidly relieve symptoms, and their use is recommended by ECCO in patients with acute bowel obstruction not requiring emergency surgery [133], but they do not grant long-term efficacy [134]. Similarly, the role of thiopurines is limited, while biologics appear more effective [134]. However, the rapid tissue healing induced by anti-inflammatory therapies tends to produce scarring phenomena. Therefore, the role of these drugs, in particular anti-TNFα, has been questioned by some authors, who described bowel strictures as a complication following treatment with infliximab [135,136,137]. Conversely, many recent studies have overcome these concerns, and it has been observed that infliximab downregulates some proteins involved in fibrogenesis, such as basic fibroblast growth factor and vascular endothelial growth factor [138]. 

Most of the currently available evidence about biologics in stricturing CD focuses on anti-TNFα. The first promising result emerged from some monocentric studies demonstrating the improvement or regression of small bowel strictures in small groups of patients treated with infliximab [139,140]. More recently, Allocca et al. conducted a real-life study on a larger cohort of 51 CD patients with small bowel or colonic stenoses, aiming to assess the efficacy of both anti-TNFα drugs (infliximab and adalimumab) in avoiding surgery. Anti-TNFα proved successful in about two thirds of patients, with the rate of surgery being less than 40% globally (39.2% in the entire cohort, 42.1% for infliximab, and 37.5% for adalimumab) [141]. The multicentric study published in 2018 by the CREOLE group appears undoubtedly noteworthy in such a context. To the best of our knowledge, it has included the largest prospective cohort so far: 97 patients with symptomatic small bowel strictures were enrolled and received subcutaneous adalimumab as per standard protocol. The primary endpoint was clinical success at 24 weeks after treatment induction, defined as adalimumab continuation without the need for corticosteroids, other biologics, endoscopic dilatation, or surgery, and without severe adverse events. About two-thirds of the patients (64%) achieved success at week 24, and about half of them were surgery-free after 4 years [142]. Furthermore, Schulberg et al. recently published the first randomized controlled trial (STRIDENT) investigating the efficacy of adalimumab in stricturing CD. More specifically, 77 patients were enrolled and randomized to receive standard adalimumab therapy (160 mg at week 0, 80 mg at week 2, then 40 mg every 2 weeks) vs. an intensive protocol (thiopurine plus adalimumab 160 mg once per week for 4 weeks, followed by 40 mg every 2 weeks). Adalimumab proved to be effective in both groups, with better clinical and radiological responses at 12 months after intensive care therapy [143]. Data from real-life analysis confirm good success rates of anti-TNFα therapy in the short term, with treatment responses decreasing over time (about 69% at 1 year, 51% at 2 years, and 28% at 5 years) [144]. The early introduction of therapy, within 18 months after CD diagnosis is associated with higher effectiveness [145].

Unfortunately, data about other biological drugs (vedolizumab and ustekinumab) are scarce and controversial. Globally, stricturing CD is less likely to respond to these therapies, although some cases of successful double therapy with vedolizumab plus ustekinumab have been described [146,147,148].

### 3.2. Intralesional Medical Therapies

Endoscopic dilatation is a recommended and effective option for short strictures under 5 cm [8]. Intralesional injection of anti-inflammatory therapies during the endoscopic procedure has been attempted as a strategy to boost the efficacy of balloon dilatation and reduce restenosis occurrence. 

Steroids are the most commonly investigated drugs for this purpose, especially triamcinolone for its long-lasting effect, and they are locally active for up to one month. However, their efficacy is still controversial, mainly due to small sample sizes and large heterogeneity in the choice of the steroid, patients’ characteristics, and endoscopic techniques [149]. Some encouraging data from case series reporting steroid injection after balloon dilatation has appeared since 1995 [150,151,152], and the first retrospective study by Singh et al. confirmed that steroid-treated patients had a lower stenosis recurrence rate than balloon dilation alone (10% vs. 31.1%) [153]. As far as we know, to date, there are only three randomized, controlled studies on the topic. In 2007, East et al. first compared intrastricture steroid injection vs. a placebo in a small group of 13 CD patients presenting ileocolonic anastomotic stenoses. In contrast with previous data, they did not find any significant improvements after steroid administration [154]. Di Nardo et al. conducted a similar study on 29 pediatric patients with anastomotic or de novo strictures, observing higher rates of re-dilation and surgery in the placebo group [155]. Both of these studies used triamcinolone, while in a recent Polish trial by Feleshtynsky et al., 64 patients were randomized to receive endoscopic dilatation with or without prednisolone injection. According to their results, local steroid treatment reduced the recurrence rate at one year from 34.4% to 9.3% [156]. Larger and more homogeneous randomized controlled trials are needed to allow the use of intralesional steroids in routine clinical practice. 

Intralesional injection of infliximab has also been attempted, in cases of rectal [157], colonic [158], and small bowel strictures [159], to directly deliver a high dose of anti-TNFα into the affected tissue. This procedure seems to be effective, leading to rapid relief of symptoms and endoscopic improvement, and it may be repeated with a good safety profile. However, the results are preliminary and are only derived from small case series, while randomized controlled trials are lacking. 

### 3.3. Novel Potential Medical Therapies

Knowledge of the etiopathogenesis of fibrosis represents the basis for developing novel therapies. Based on the pathogenetic mechanisms described in the previous sections, researchers are examining new therapeutic agents, both through pre-clinical models and in some pivotal clinical trials on human beings. Most of the available data refer to pro-fibrotic cytokines/soluble factors and to the modulation of ECM production; moreover, some recent evidence focuses on the microbiota. Studies on potential therapies acting at the genetic or epigenetic level, as well as drugs inhibiting creeping fat formation, are still lacking. 

Due to its central role in fibrogenesis, many studies have naturally investigated TGF-β signaling. Thus, Imai et al. examined mice with 2,4,6-trinitrobenzene sulfonic acid (TNBS)-induced colitis and treated them with TM5275, an inhibitor of plasminogen activator inhibitor-1 (PAI-1), which is a downstream target of the TGF-β pathway, preventing enzymatic degradation of the ECM. The oral administration of TM5275 leads to an up-regulation of MMP-9 and subsequent decreased collagen accumulation [160]. Furthermore, small molecules inhibiting Wnt signaling also proved to reduce collagen and block the TGF-β, cascade both in human intestinal fibroblast CCD-18Co cell lines and in myofibroblasts isolated from surgically resected fibrotic specimens [161]. Interestingly, the oriental traditional herbal medicine Daikenchuto also revealed beneficial effects against intestinal fibrosis in animal models by activating the transient receptor potential ankyrin 1 (TRPA1) channel in intestinal myofibroblasts. After one week of rectal administration through enemas, Daikenchuto was shown to counteract TGF-β effects, reducing the expression of type I collagen and α-SMA [162]. Among the other soluble factors, the aforementioned study by Scheibe et al. assessed the efficacy of antibodies against IL-36R on mouse models of colitis, obtaining a reduction in the established fibrosis [84]. 

ECM production and degradation represent another promising target, so different authors have investigated the effect of therapies on regulating fibroblasts/myofibroblasts activation. Yang et al. observed that microvesicles containing mi-RNA-200b are able to suppress EMT in intestinal epithelial cells in vitro by targeting the pro-fibrotic proteins Zinc finger E-box binding homeobox 1 and 2 (ZEB1 and ZEB2) [163]. Truffi et al. collected stenotic and non-stenotic ileal specimens from patients undergoing surgical resection for stricturing CD and established bowel tissue and myofibroblast ex vivo cultures. They observed an overexpression of the fibroblast activation protein (FAP) in the stenotic tracts compared with non-stenotic ones and demonstrated that treatment with anti-FAP antibodies induced a reduction in collagen production in a dose-dependent manner [164]. Moreover, the influence of mesenchymal stem cells on regulating myofibroblasts’ activity has been described. In an in vitro study by Choi et al., human myofibroblasts were co-cultured with stem cells deriving from perinatal tissue (umbilical cord and placenta). The authors observed that stem cells blocked the TGF-β1-induced pathway of Rho/MRTF/SRF (Ras homolog/myocardin-related transcription factor/serum response factor). This led to a reduced expression of procollagen1A1, fibronectin, and α-SMA in myofibroblasts, resulting in downregulated fibrogenesis, thus suggesting a novel therapeutic candidate [165]. Indeed, the efficacy of stem cells has been further assessed in a recent phase I-II pilot trial on 10 CD patients who received intralesional treatment with allogenic bone marrow stem cells during colonoscopy. Stem cell injection proved to be a safe and well tolerated procedure, but a complete or partial resolution of the stenosis was observed only in half of the patients, and several occlusions occurred in the follow-up period. The small sample size and the lack of a control group do not allow a reliable evaluation of the role of such treatment [166]. 

The use of some probiotics has also been attempted to reduce fibrosis by modulating the gut microbiota. Liu et al. have identified a new probiotic strain with anti-inflammatory properties, *Lactococcus lactis* ML2018, demonstrating its influence in ameliorating DSS-induced colitis and controlling fibrosis [167]. In a study conducted by Park et al., mice with DSS-induced colitis were orally administered a complex of 12 probiotics in combination with prebiotics (chicory fiber), rosavin, and zinc. In addition to a beneficial effect on intestinal inflammation, the authors also described a decrease in α-SMA and type I collagen levels [168]. Similarly, other probiotic combinations (mainly based on Lactobacilli and Bifidobacteria) have been tested on murine models or human colonic fibroblasts, with encouraging results on fibrosis [169,170]. Interestingly, Kashima et al. tested a bacterial-derived molecule, polyphosphate, from *Lactobacillus brevis*, finding reduced levels of TGF-β1, TNFα, and IL-1β after treatment, resulting in a significant improvement in inflammation and fibrosis [171]. 

Table 2 provides a schematic overview of the main studies published on new potential antifibrotic treatments, highlighting their link with specific pathogenetic mechanisms.

### 3.4. Ongoing Clinical Trials

To date, some clinical trials are ongoing, testing the efficacy of approved or new pharmacological therapies for fibrostenotic CD. In particular, among the novel potential therapies, one study (NCT02675153) is investigating the role of rapamycin, an allosteric inhibitor of mTOR complex 1 [172], while another (NCT05013385) is examining spesolimab, a monoclonal antibody against the IL-36 receptor, which is already approved for psoriasis [174]. The main characteristics of ongoing clinical trials are summarized in Table 3, and further details are available online at clinicaltrials.gov (https://clinicaltrials.gov/ accessed on 15 March 2024).

If targeting intestinal fibrosis still represents a hard challenge, recent progress in understanding its pathogenesis has paved the way for new, promising therapeutic options. The identification of some predisposing genetic and epigenetic factors does not provide immediate treatment applications yet, but may help to recognize high-risk patients who could benefit from a stricter follow-up and a personalized approach. On the other hand, the considerable progress concerning molecular pathways contributing to fibrogenesis has led to the development of new therapeutic agents, especially targeting TGF-β or ILs, ECM balance, and, more recently, the gut microbiota. The results are encouraging, but most of them rely on pre-clinical studies, including in vitro or in vivo models. Moreover, most murine models are mice with chemically induced colitis, while CD inflammation may affect any segment of the GI tract, and stenoses are more frequent in the ileum. Data about human beings, especially the small bowel, are still lacking. Future research should try to fill such gaps as well as design studies comparing pharmacological treatments for fibrosis with surgery or endoscopy, the current gold standard for CD strictures. Finally, no evidence is available about potential interactions between the proposed anti-fibrotic therapies and the anti-inflammatory therapies that are currently approved. Further studies are necessary to establish how to incorporate the novel agents into the IBD therapeutic algorithm.

## 4. Conclusions

Medical treatment of stricturing CD is an urgent challenge for IBD physicians, as it is still encumbered by many shadows. The currently available drugs have limited impact on intestinal fibrosis, and surgical or endoscopic interventions are often required in this class of patients. However, in the last few years, increased knowledge on the molecular mechanisms underlying fibrostenosing CD has helped hypothesize new potential pharmacological targets, and some encouraging results have begun appearing in the literature. Interventional trials on CD patients are definitely needed to develop novel treatments and transform results from basic science into clinical practice. 

## Figures and Tables

**Figure 1 ijms-25-06326-f001:**
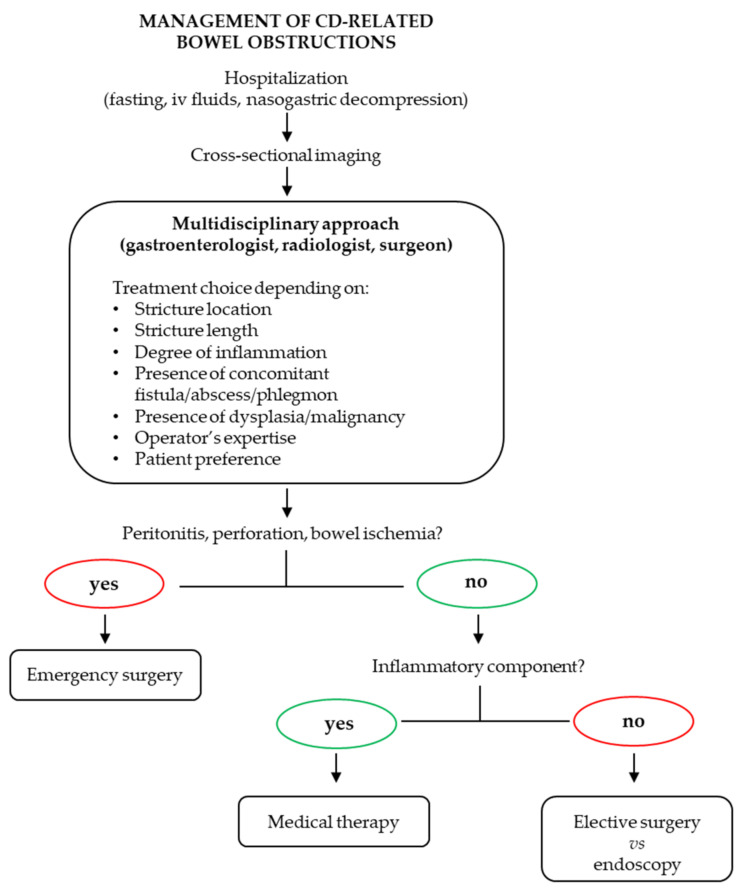
Therapeutic algorithm for bowel obstruction in CD. A multidisciplinary approach is crucial to choose the best therapeutic option. The role of pharmacological therapy is limited to bowel strictures with an inflammatory component, while surgery is required both in emergent complicated situations and in cases of stenoses showing a prevalent fibrostenotic pattern.

**Figure 2 ijms-25-06326-f002:**
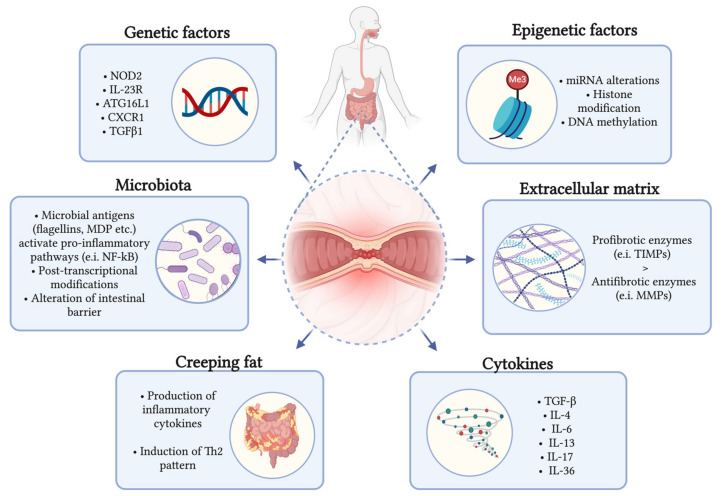
Pathogenetic factors influencing intestinal fibrosis development in fibrostenosing CD. Abbreviations: NOD2: nucleotide-binding oligomerization domain-containing gene 2; IL-23R: interleukin-23 receptor; ATG16L1: autophagy-related 16-like 1 gene; CX3CR1: C-X3-C Motif Chemokine Receptor 1; TGFβ1: transforming growth factor beta 1; miRNA: micro-ribonucleic acid; DNA: deoxyribonucleic acid; MDP: muramyl dipeptide; NF-kB: nuclear factor kappa-light-chain-enhancer of activated B cells; TIMPs: tissue inhibitors of metalloproteinases; MMPs: matrix metalloproteinases; Th: T-helper cells.

**Figure 3 ijms-25-06326-f003:**
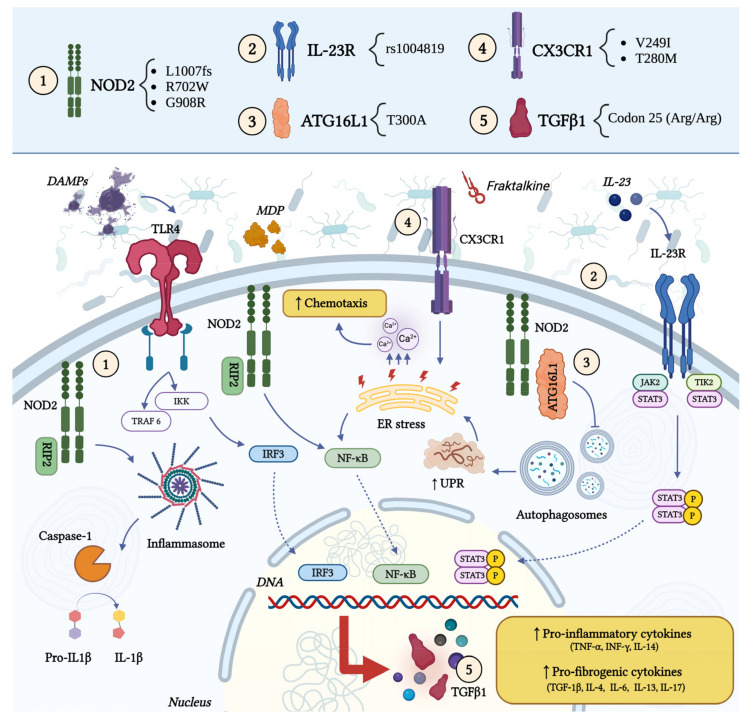
Genetic factors influencing the pathogenesis of stenoses in CD and their corresponding molecular pathways. Abbreviations: ATG16L1: autophagy-related 16-like 1 gene; CX3CR1: C-X3-C Motif Chemokine Receptor 1; DAMPs: damage-associated molecular patterns; DNA: deoxyribonucleic acid; ER: endoplasmic reticulum; IL: interleukin; IL-R: interleukin receptor; IKK: IkappaB kinase; IFN-γ: interferon gamma; IRF3: interferon regulatory factor 3; JAK2: Janus Kinase 2; MDP: muramyl dipeptide; NOD2: nucleotide-binding oligomerization domain-containing gene 2; NF-kB: nuclear factor kappa-light-chain-enhancer of activated B cells; P: phosphate group; RIP2: receptor-interacting protein kinase 2; STAT3: signal transducer and activator of transcription 3; TGFβ1: transforming growth factor beta 1; TLR4: Toll-like receptor 4; TNF α: tumor necrosis factor alpha; TRAF6: tumor necrosis factor receptor associated factor 6; UPR: unfolded protein response.

**Table 1 ijms-25-06326-t001:** Main cytokines and soluble factors involved in fibrogenesis in Crohn’s disease.

**Cytokines** **Soluble Factors**	**Mechanism**	**Activity**
TGF-β	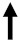	SMAD pathway → ECM deposition.	PROFIBROTIC
IL-6	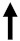	Cell migration-inducing hyaluronan-binding protein → hyaluronan degradation.
IL-36	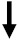	Expression of the metalloproteinase MMP13.
IL-4 IL-5 IL 13	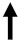	Th2 pathway → ECM deposition.
IL-17	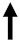	Th17 pathway → ECM deposition.
mTOR complex	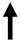	Synthesis of collagen, fibroblasts, and myofibroblast.Production of profibrogenic cytokines (IL-4, IL-6, IL-13, IL-17, and TGF-β1).	
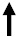
IFN-γ	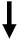	Th1 pathway → ECM deposition.	ANTIFIBROTIC

TGF-β: Transforming Growth Factor-β; IL: interleukin; mTOR: mTOR: mammalian target of rapamycin; IFN: interferon; SMAD: small mother against decapentaplegic; ECM: extracellular matrix; Th: T helper cells.

**Table 2 ijms-25-06326-t002:** New potential therapeutic agents for intestinal fibrosis and their molecular mechanisms. Further details are provided in the main text.

Cytokines/Soluble Factors		ECM		Microbiota	
Therapeutic Agent	Target Pathway	Therapeutic Agent	Target Pathway	Therapeutic Agent	Target Pathway
TM5275 [160]	TGF-β signaling; inhibition of PAI-1	Anti-FAP antibodies [164]	Inhibition of FAP; decrease in type-I collagen; reduced TIMP-1 levels	Polyphosphate [171]	Reduced levels of TGF-β1, TNFα and IL-1β
3235-0367 Wnt-C59 ICG-001 [161]	TGF-β signaling; inhibition of Wnt signaling	Perinatal stem cells (umbilical/placenta) [165]	Inhibition of RhoA, MRTF-A, and SRF expression;reduced expression of procollagen1A1, fibronectin, and α-SMA	12 probiotics, prebiotics, rosavin, and zinc [168]	Decrease in α-SMA and type I collagen
Daikenchuto [162]	TGF-β signaling; activation of TRPA1; reduced production of type I collagen and α-SMA	Allogenic bone marrow stem cells [166]	NK	*Lactococcus lactis* ML2018 [167]	
Rapamycin [172]	Inhibition of mTOR complex	Adipose-derived allogenic mesenchymal stem cells [173]	NK	*Lactobacillus plantarum*, *L. acidophilus*, *L. rhamnosus*, and *Bifidobacterium animalis* [169]	TGF-β1/Smad signalling
Spesolimab (monoclonal antibody) [174]	Inhibition of IL-36 receptor	Microvesicles containing mi-RNA-200b [163]	Inhibition of ZEB1 and ZEB2; inhibition of EMT	Multi-Strain Probiotic Formulation (Vivomixx^®^) [170]	TGF-β1/Smad signalling
Anti-IL-36R antibodies [84]	Inhibition of IL-36 receptor				

TGF-β: Transforming Growth Factor-β; PAI: plasminogen activator inhibitor; Wnt: Wingless-Int-1; TRPA1: transient receptor potential ankyrin 1; α-SMA: α-smooth muscle actin; mTOR: mammalian target of rapamycin; IL: interleukin; FAP: fibroblast activation protein; miRNA: micro-ribonucleic acid; TIMP: tissue inhibitor of metalloproteinases; Rho: Ras homolog; MRTF: myocardin-related transcription factor; SRF: serum response factor; NK: not known; ZEB: Zinc finger E-box binding homeobox; EMT: epithelial-mesenchymal transition; TNF: tumor necrosis factor; Smad: small mother against decapentaplegic.

**Table 3 ijms-25-06326-t003:** Ongoing clinical trials on medical therapies for fibrostenosing CD.

Title	Study ID	Study Type	Phase	Drug	Arms	Randomization	Placebo	Primary Outcome	Status
A Randomized, Double-blinded, Placebo-controlled Study on the Effects of Adalimumab Intralesional Intestinal Strictures of Crohn’s Disease Patients [173]	NCT01986127	Interventional	III	Adalimumab (intralesional administration during endoscopy)	Two arms: dilatation and adalimumab vs. dilatation and placebo	Yes	Yes	Successful dilatation at week 8	Concluded
SyMptomAtic Stricturing Small Bowel CRohn’s Disease—Medical Treatment Versus Surgery, a Prospective, Multi-centre, Randomized, Non-inferiority Trial [175]	NCT05584228	Interventional	NA	Azathioprine per os and infliximab sc	Two arms: medical therapy vs. surgery	Yes	Yes	Clinical: IBD-related quality of life at 12 months	Not yet recruiting
Efficacy of Ustekinumab-based Integrated Medicine Therapy in Patients With Symptomatic Stricturing Crohn’s Disease: a Multicentre, Prospective, Observational Cohort Study [176]	NCT05387031	Observational	NA	Ustekinumab	Single arm	NA	NA	Treatment success at week 52	Recruiting
Surgical Intervention Versus Biologics Treatment for Symptomatic Stricturing Crohn’s Disease (SIBTC): an Open-label, Single-center, Randomized Controlled Trial [177]	NCT05421455	Interventional	NA	Biologics	Two arms: (up to two) biologics vs. surgery	Yes	No	Clinical remission rate at 1 year	Recruiting
Efficacy and Safety of Rapamycin in the Treatment of Crohn’s Disease-related Stricture [172]	NCT02675153	Interventional	NA	Rapamycin 2 mg/day for six months	Single arm	No	No (open label)	Response rate at 24 weeks (ability to tolerate normal diet; need for endoscopy or surgery; adverse events)	Unknown
Multi-center, Double-blind, Randomized, Placebo-controlled, Phase IIa Trial to Evaluate Spesolimab (BI 655130) Efficacy in Patients With Fibrostenotic Crohn’s Disease [174]	NCT05013385	Interventional	IIa	Spesolimab	Two arms: spesolimab vs. placebo	Yes	Yes	Clinical: proportion of patients with symptomatic strictures at week 48. Radiological: Proportion of patients with radiographic stenosis response at week 48.	Concluded
Clinical Trial Phase IIa to Evaluate the Safety and Effectiveness of Treatment With Fat-derived Mesenchymal Allogenic Mesenchymal Troncal Cells in Patients With Single Inflammatory Stenosis in the Context of Crohn’s Disease [178]	NCT05521672	Interventional	IIa	Adipose-derived allogenic mesenchymal stem cells (perilesional injection during laparoscopic procedure)	Single arm	No	No	Safety. Radiological improvement in stenoses. Clinical (CDAI and IBDQ). Need for surgery.	Recruiting

NA: not applicable; CDAI: Crohn disease activity index; IBDQ: Inflammatory Bowel Disease Questionnaire Open issues and future perspectives.

## Data Availability

No new data were created or analyzed in this study. Data sharing is not applicable to this article.

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
