# Peer review of "Fibrostenosing Crohn’s Disease: Pathogenetic Mechanisms and New Therapeutic Horizons"

_ijms, 2024, doi:10.3390/ijms25126326_

Round 1

Reviewer 1 Report

Comments and Suggestions for Authors

In this review, authors aim to summarize the molecular factors involved in intestinal fibrosis’ pathogenesis and its therapies. The review is well-organized and the topic is interesting. There are some suggestions:

1. In the Abstract, it is suggested that authors should include more information about the therapy.

2. In the Cytokines and other soluble factors, can authors summarize them into a table or a figure?

3. The figure quality should be improved.

Author Response

In this review, authors aim to summarize the molecular factors involved in intestinal fibrosis’ pathogenesis and its therapies. The review is well-organized and the topic is interesting. There are some suggestions:

  1. In the Abstract, it is suggested that authors should include more information about the therapy.

R: We appreciated your note and have revised the abstract by adding more specific details on therapeutic approaches.

  1. In the Cytokines and other soluble factors, can authors summarize them into a table or a figure?

R: following reviewer’s suggestion, we have added a summarizing table at the end of the specific subsection about cytokines and soluble factors.

  1. The figure quality should be improved.

R: thank you for your attention in reviewing our images. We have now re-uploaded the images improving their resolution, as it is in their original format (600DPI).

Reviewer 2 Report

Comments and Suggestions for Authors

I find this to be a very interesting topic, but I do not understand the methodology followed for the review. At no point is the search strategy specified, nor are the keywords used for the search. There is also no indication of which databases were searched. Therefore, I believe this article may be biased. A comprehensive systematic review could have been conducted for it to be publishable. However, the information provided is not valid as a summary of the literature if a guideline for searching that information, such as PRISMA 2020, has not been followed.

Author Response

I find this to be a very interesting topic, but I do not understand the methodology followed for the review. At no point is the search strategy specified, nor are the keywords used for the search. There is also no indication of which databases were searched. Therefore, I believe this article may be biased. A comprehensive systematic review could have been conducted for it to be publishable. However, the information provided is not valid as a summary of the literature if a guideline for searching that information, such as PRISMA 2020, has not been followed.

R: the aim of our work is to provide a comprehensive overview of current knowledge about the pathogenesis and treatment of fibrostenosing Crohn’s disease. However, this is a narrative, not a systematic review, reason explaining the lacking description of a structured search strategy.

Reviewer 3 Report

Comments and Suggestions for Authors

Dear Authors,

 I have carefully reviewed your manuscript on the pathogenesis of fibrostenosing Crohn's disease and would like to provide feedback on its strengths and areas for improvement.

 Strengths:

1. The visual aids, particularly the figures, are well-crafted and enhance the understanding of the content. They are intuitive and easy to comprehend, aiding in the communication of complex concepts.

2. Your in-depth exploration of the pathogenesis of fibrostenosing Crohn's disease is commendable. The detailed analysis of the molecular mechanisms involved in intestinal fibrosis provides a comprehensive insight into the disease, contributing to a deeper understanding of its complexities.

3. The extensive referencing of relevant literature in the field adds credibility to your work. By drawing on a wide range of sources, you have established a solid foundation for understanding the pathogenesis and treatment of Crohn's disease.

 Areas for Improvement:

1. Some of the references cited in the manuscript appear to be dated. It would be beneficial to include more recent studies to ensure the incorporation of the latest advancements and findings in the field.

2. There is a minor error in the manuscript where "Epigenetic factors" on line 236 should be corrected to "2.2 Epigenetic factors" for clarity and consistency in the text.

 Overall, your manuscript presents a valuable contribution to the understanding of fibrostenosing Crohn's disease. Addressing the mentioned areas for improvement will further enhance the quality and accuracy of your work.

 Thank you for your dedication to this important research topic, and I look forward to seeing the revised version of your manuscript.

 Best regards,

Comments on the Quality of English Language

Minor editing of English language required

Author Response

Dear Authors,

I have carefully reviewed your manuscript on the pathogenesis of fibrostenosing Crohn's disease and would like to provide feedback on its strengths and areas for improvement.

 Strengths:

  1. The visual aids, particularly the figures, are well-crafted and enhance the understanding of the content. They are intuitive and easy to comprehend, aiding in the communication of complex concepts.
  2. Your in-depth exploration of the pathogenesis of fibrostenosing Crohn's disease is commendable. The detailed analysis of the molecular mechanisms involved in intestinal fibrosis provides a comprehensive insight into the disease, contributing to a deeper understanding of its complexities.
  3. The extensive referencing of relevant literature in the field adds credibility to your work. By drawing on a wide range of sources, you have established a solid foundation for understanding the pathogenesis and treatment of Crohn's disease.

 Areas for Improvement:

  1. Some of the references cited in the manuscript appear to be dated. It would be beneficial to include more recent studies to ensure the incorporation of the latest advancements and findings in the field.

R: thank you for your suggestion. We have updated the bibliography adding some more recent references.

  1. There is a minor error in the manuscript where "Epigenetic factors" on line 236 should be corrected to "2.2 Epigenetic factors" for clarity and consistency in the text.

R: Thank you for noticing that error. We have corrected it.

Overall, your manuscript presents a valuable contribution to the understanding of fibrostenosing Crohn's disease. Addressing the mentioned areas for improvement will further enhance the quality and accuracy of your work.

Thank you for your dedication to this important research topic, and I look forward to seeing the revised version of your manuscript.

Round 2

Reviewer 2 Report

Comments and Suggestions for Authors

I understand what a narrative review is, but in my opinion, these reviews are of lower rank on the scale of scientific articles. This is because the authors may be selecting articles and information randomly, potentially introducing bias to the real, objective, and reproducible information that a systematic review on the same topic would provide. I encourage you to undertake a systematic review instead, as a narrative review does not seem to offer a significant contribution to the field of this disease.

Author Response

Following reviewer’s and editor’s suggestions, we have declared in the introduction the narrative nature of our review, specifying the associated limitations. We have also emphasized that the section about new therapies represents a general overview of the most important progresses, while the main focus of our work is about molecular bases of stricturing Crohn’s disease, providing a general framework for future research about their therapeutic applications.